

# The concurrence of Atmospheric Rivers and explosive cyclogenesis in the North Atlantic and North Pacific basins

Jorge Eiras-Barca[1], Alexandre M. Ramos[2*], Joaquim G. Pinto[3], Ricardo M. Trigo[2], Margarida L.R. Liberato[2,4], and Gonzalo Miguez-Macho[1]

[1] Non-Linear Physics Group, Universidade de Santiago de Compostela, Galicia, Spain
[2] Instituto Dom Luiz, Faculdade de Ciências, Universidade de Lisboa, Lisbon, Portugal;
[3] Institute for Meteorology and Climate Research (IMK-TRO), Karlsruhe Institute of Technology (KIT), Karlsruhe, Germany
[4] Escola de Ciências e Tecnologia, Universidade de Trás-os-Montes e Alto Douro, Vila Real, Portugal.

*Correspondence to*: A. M. Ramos (amramos@fc.ul.pt); and and J.Eiras-Barca (jorge.eiras@usc.es).

**Abstract.** The explosive cyclogenesis of extra-tropical cyclones and the occurrence of atmospheric rivers

are characteristic features of baroclinic atmospheres, and are both closely related to extreme hydrometeorological events in the mid-latitudes, particularly on coastal areas on the western side of the continents. The potential role of atmospheric rivers in the explosive cyclone deepening has been previously analysed for selected case studies, but a general assessment from the climatological perspective is still missing. Using ERA-Interim reanalysis data for 1979-2011

, we analyse the concurrence of atmospheric rivers and explosive cyclogenesis over the North Atlantic and North Pacific Basins for the extended winter months (ONDJFM). Atmospheric rivers are identified for almost 80% of explosive deepening cyclones.  For non-explosive cyclones, atmospheric rivers are found only in roughly 40% of the cases. The analysis of the time evolution of the high values of water vapour flux associated with the atmospheric river during the cyclone development phase leads us to

hypothesize that the identified relationship is the fingerprint of a mechanism that raises the odds of an explosive cyclogenesis occurrence and not merely a statistical relationship. This insight can be helpful for the predictability of high impact weather associated with explosive cyclones and atmospheric rivers.




# 1 Introduction

Intense extra-tropical cyclones are one of the major natural threats in mid-latitudes and are often
5  responsible for large socioeconomic impacts (Munich Re, 2015). Their impacts include strong winds,
heavy precipitation and in some cases storm surges (e.g., Lamb, 1991) In particular, cases associated with
explosive cyclogenesis (Sanders and Gyakum, 1980; pressure decrease larger than 24 hPa in 24 hours at
60ºN, or equivalent) are associated with particularly large impacts and often with low predictability (e.g.
Wernli et al., 2002; Fink et al., 2009). Such systems are often referred in the literature as "bombs".
10  According to Shapiro et al. (1999), explosive cyclogenesis result from different mechanisms that include
upper-level cyclonic vorticity advection, low-level warm air advection, and latent heat release. This may
be supported by upper-tropospheric Rossby wave breaking, which constraints and intensifies the upper-
level jet stream and thus contributes to intense cyclone developments (e.g. Hanley and Caballero, 2012;
Gómara et al., 2014). According to Aubert (1956), the latent heating influence is significant not only in
15  the pressure distribution, but also in the vertical motion field. In particular, Aubert (1956) states that this
mechanism lowers the heights of isobaric surfaces in the lower troposphere, and raises them in the upper.
In agreement, Tsou et al. (1987) found that even for cases with strong vorticity advection and the
differential thermal advection, that latent heat release is still primary cause of the pressure falling below
900 hPa during the period of most abundant precipitation. The release of latent heat contributes to increase
20  the storm's available potential energy, deepening the cyclone while decreasing the horizontal scale of the
region of ascent particularly in the most maritime cyclones (e.g., Emanuel, 1987; Snyder and Lindzen,
1991; Davis, 1992)

Several studies have confirmed the occurrence of a maximum of latent and sensible heat
availability in the lower troposphere near the warm sector of the cyclone, and documented the contribution
25  of moist diabatic processes such as latent heat release by cloud condensation processes to the
intensification of extratropical cyclones (Pinto et al., 2009, Liberato et al., 2013; Ludwig et al., 2014).
However, the relative contribution of diabatic processes to the cyclone deepening may differ considerably
from case to case. For example, Fink et al. (2012) showed for selected explosive cyclogenesis cases that
while diabatic processes played a key role for storms such as Xynthia and Klaus, other storms are largely



baroclinic driven (e.g. Kyrill and Martin). The recent study by Pirret et al. (2017) quantifies the role of the different mechanisms for the deepening of 60 severe European windstorms based on the methodology by Fink et al. (2012). This objective assessment shows that baroclinic processes dominate the majority of storms, such that deepening is closely related to warm advection ahead of the cyclone centre.

Contributions from diabatic processes vary strongly from cases to case and exceed those from horizontal temperature advection in 10 out of 58 cases, with values of up to 60%. In addition, the diabatic contribution is significantly correlated with the period of time a given storm spends on the equatorward side of the jet, where there is greater potential for diabatic processes in the warm, moist air.

The higher moisture availability in the north Pacific and north Atlantic basins are controlled by
so-called Atmospheric Rivers (AR; e.g. Newell and Zhu, 1994; Zhu and Newell 1998; Bao et al., 2006; Ralph and Dettinger, 2011; Gimeno et al., 2017). ARs are relatively narrow (on average 500 km) corridors of enhanced water vapour (WV) transport in the lower troposphere that can extend for thousands of kilometers. This phenomenon is associated with tropical moisture exports (TME; e.g., Knippertz and Wernli, 2010) and occurs often in combination with the passage of extratropical cyclones (Dacre et al.,
2015). Recently, some agreement has been achieved (Dettinger et al., 2015) regarding the relationships between ARs, warm conveyor belts (WCBs), and tropical moisture exports (TMEs). The term WCB refers to the zone of dynamically uplifted heat and vapour transport close to a mid-latitude cyclone. This vapour is often transported to the WCB by an AR, and the result of the uplift is heavy rainfall that generally marks the downwind end of an AR, provided that the AR has not experienced orographic uplift (upslope
flow), accompanied by rainout over mountains earlier on its approach to the WCB. TMEs are zones of intense moisture transport out of the tropics, vapour that is frequently conducted by ARs towards cyclones and WCBs. TMEs can provide important vapour sources for ARs, but most ARs also incorporate mid-latitude sources and convergences of vapour along their paths (Dettinger et al., 2015; Sodemann and Stohl, 2013, Dacre et al., 2015). A comprehensive assessment of moisture sources feeding Atlantic ARs
bound for different sectors of Western Europe can be found in Ramos et al. (2016). The importance of ARs in extreme precipitation events and floods has been analysed in detail for the west coast of the USA (particularly for California) over the last decade (e.g., Ralph et al., 2004; Neiman et al., 2008; Dettinger et al., 2011). Similar conclusions have been reached for Europe (e.g. Lavers et al., 2012; Liberato et al.





2013; Ramos et al., 2015; Eiras-Barca et al., 2016; Brands et al., 2017) and other regions of the world (e.g. Viale and Nuñez, 2011; Mahoney et al., 2016; Blamey et al., 2017).

Given the role of latent heat release in the development of explosive cyclogenesis, this suggests that explosive cyclogenesis in the mid-latitudes may be influenced by the presence of an AR. Additionally, the release of sensible heat in the vicinity of the cyclone will enhance the convective instability of the AR. Previous studies showed for selected case studies that explosive development can indeed be driven by the presence of an AR (Zhu and Newell, 1994, Ferreira et al., 2016). For example, the Ferreira et al (2016) have provided evidence on the role of a TME over the western and central (sub)tropical Atlantic towards cyclone development, which converged into the cyclogenesis region and then moved along with the storm towards Europe. However, to the best of our knowledge, no ample assessments analysing the role of ARs in the explosive deepening have been performed from the climatological perspective. The main objective of this study is to provide a comprehensive evaluation on the role of the ARs in the explosive deepening of North Atlantic and North Pacific extratropical cyclones between 1979 and 2011 for the extended winter months (ONDJFM), focusing on the spatial concurrence and the timing of both features.

The manuscript is organized as follows: Section 2 described the data and methods, while the results are presented in section 3. Finally, the conclusions are given and discussed in section 4.

## 2 Data and Methods

We use ECMWF ERA-Interim Reanalysis (Dee et al., 2011) between 1979 and 2011 for our study. For the cyclone detecting and tracking methodology (see section 2.1), 6-hourly instantaneous 1000 hPa geopotential height fields at a resolution of 0.75º × 0.75º are considered. For the detecting and tracking of the ARs structures (see section 2.2), we used moisture and wind values at multiple vertical pressure levels to compute the integrated water vapour column (IWV) and the vertically integrated horizontal water vapour transport (IVT) at the same resolution.



### 2.1 Cyclone Detecting and Tracking Methodology

We have applied an automatic procedure to identify and track extratropical cyclones (Trigo, 2006). This particular cyclone detecting and tracking algorithm was first developed for the Mediterranean region (Trigo et al., 1999, 2002), later extended to a larger Euro-Atlantic region (Trigo, 2006) and finally

generalized for both hemispheres (e.g. Neu et al. 2013). The scheme is applied to the ERA-Interim geopotential height at 1000 hPa (Z1000) fields. Cyclones are identified and tracked at a 6-hourly basis at the spatial resolution available of $0:75º \times 0:75º$ for the entire northern hemisphere. Results from this method compare well with other similar methods (Neu et al., 2013). Storms with minimum central pressure higher than 1010 hPa over their entire lifecycle and lasting more than 24 hours are discarded

from the subsequent analysis. For each cyclone, the maximum deepening rate $\Delta P$ per cyclone track is determined by the maximum pressure drop at the centre of the cyclone on the basis of all the 6-h successive time steps $((P(t-6)-P(t))$ in its life cycle. The maximum deepening point (MDP) corresponds to the point P(t) and is attained for each cyclone in order to analyse the influence of the ARs on its maximum deepening rate. We choose the maximum deepening point over the minimum pressure

point of the cyclone because the minimum pressure point may not guarantee that trigger for its deepening occurred in the 6 hours previous to it. On the contrary, by choosing in the MDP, we guarantee that the trigger effect for its maximum deepening occurred just prior to it. From this database, the sub-set of explosive cyclogenesis (EC) is selected for further analysis. Following Bergeron (1954) and the generalization by Sanders and Gyakum (1980), explosive cyclones are defined as cyclones with

deepening rates exceeding 24 hPa in 24 hours for a reference latitude of 60°N $(24 * \sin(\varphi) = \sin(60º))$ hPa / 24 hours, where $\varphi$ is the latitude of the cyclone core at MDP. All remaining cyclones are included in the sub-set non-explosive cyclones (NEC). In addition, for the ARs analysis, all the cyclones below 25ºN were filtered in order to avoid tropical storms and hurricanes in our analysis.

Two wide domains over both ocean basins have been selected for latitudes between 25ºN and

65ºN. For the Atlantic domain, longitudes between 80ºW and 10ºE are considered, while for the Pacific domain longitudes are between 120ºE and 105ºW. Regarding the extended winter months, For the North Atlantic basin, a total of 7315 cyclones were detected, from which 733 were classified as EC (close to



10%). Regarding the Pacific domain, a total of 10890 cyclones where identified, in which 1115 classified as EC (close to 10.3%).

Fig. 1 shows the spatial distribution of the positions where EC reached their minimum core pressure during lifetime. Depicted are the number of events per extended winter (ONDJFM) season per
$5°x5°$ grid box, normalized to the corresponding area for $50°N$ (about $200x10^3$ km$^2$). The density maps of cyclone positions provide a good overview of the distribution of explosive cyclones in both basins. Whereas for the Atlantic storm track has a clear SW-NE orientation is found, reaching values of 0.8 events per extended winter near the American continent, over the Pacific Basin the storm track is more zonal and reached values of 0.9 events per extended winter over the central-western basin.

### 2.2 Atmospheric Rivers Detection

There are several methodologies to the detect ARs, which can be broadly be divided into two groups considering the nature of the main dataset used, either satellite data or reanalysis data (Gimeno et al., 2014). For methods using satellite data, the different approaches consider the IWV, obtained mainly
from the SSM/I sensor (e.g., Ralph et al. 2004; Guan et al. 2010; Ralph and Dettinger 2011). For methods based on reanalysis data, they focus on the IVT (e.g., Zhu and Newell 1998; Lavers et al. 2013) or a combination of IWV and IVT estimated reanalysis datasets (e.g Eiras-Barca et al., 2016). An overview of the different methods to identify ARs can be found in Gimeno et al. (2014).  Given the different approaches to identify them, and in order to estimate the sensitivity of the results for the choice of
identification method, we employ two different methods to identify them.

The first method is an adaptation of Eiras-Barca et al. (2016) approach (hereafter EIRAS2016) which uses not only IWV but also the IVT to identify ARs. For each cyclone, the location and timing of the MDP (cf. section 2.1) along the cyclone track is used as a starting point. In 6 hourly time-steps and for a ±36 hour window-frame around the MDP, we search within a radius of 1500 km surrounding the
centre of the cyclone location at that time for the maximum values of IWV which are above the local 85th monthly percentile. If a grid point is selected, the neighbouring grid points are also investigated. This procedure continues as long as the threshold conditions are met, building 2-D features. A feature must have a minimum extension of 2000km to be considered an AR. The search radius of 1500 km has been



selected to take the shape and geometry of ARs and cyclones into account. While smaller radius of search may be in some cases insufficient to detect ARs in the vicinity of large cyclones, larger radius of search could detect other ARs which are unrelated to the analysed cyclone, leading to a false detection.

The second ARs detection scheme was developed by Guan and Waliser et al., (2015) (hereafter GUAN2015). The database used here (shape boundary and axis of the ARs) were provided by the authors. For this method, there is no need of a reference starting point to search for the ARs. Instead, the method isolates contiguous regions of the word of enhanced IVT exceeding a certain IVT threshold (> 85th percentile or 100kg m$^{-1}$ s$^{-1}$, whichever is greater). Each of these regions will be subsequently analysed for the geometry requirement of length >2000km, length/width ratio >2 and other considerations indicative of ARs conditions (cf. Guan and Waliser, 2015). Both algorithms operate using variable spatial and time dependent thresholds. The assignment of the cyclones to the GUAN2015 ARs is performed in an identical way as for EIRAS2016 to warrant comparability. Both methodologies were applied to both EC and NECs sub-sets in order to quantify the role of the ARs for the development of explosive cyclones.

## 2.3 Example of detection

A good example of a well-defined AR can be found in Fig.2. The selected case corresponds to an explosive cyclone where the MDP occurred on the 31 January 1988 at 18 UTC west of Ireland (approximately at latitude 51ºN and longitude 20ºW, red dot in Fig. 2b and Fig. 2c). The overall IVT pattern is clearly compatible with the presence of an AR-like structure in the North Atlantic Ocean, showing an extensive region with high IVT values extending from the Caribbean to the British Isles (Fig. 2a). This preliminary visual assessment is confirmed using the GUAN2015 algorithm in Fig. 2b, where the two highlighted regions that are distinguished corresponds to the "shape" region (reddish) which is the region where the AR can exist, and the blue line depicts the central axis of maximum intensity of the AR detected by the GUAN2015 method. Similarly, in Fig. 2c the results for the EIRAS2016, for the same case, is shown, where it shows the detection of the central axis of the AR (blue crosses) event stated in Fig. 2a.



## 3 Results

Both AR detection methodologies were applied to the entire cyclone database. The obtained information was used to estimate the relevance of the ARs in the occurrence of explosive cyclogenesis and compare it with the corresponding NEC results. First, we analyse the samples of cyclones in each

sub-set in terms of the evolution of core pressure over time. Figure S1 depicts the distributions of core pressure values from - 36 hours until +36 hours from MDP, for both the North Atlantic and North Pacific basins and for EC and NEC. Please note that the number of cyclones included in the statistics changes over time (Table S1), as not all systems have the same life time. EC systems typically deepen around 30 hPa during its lifetime and attain a minimum core pressure around MDP+6 or +12 hours. Afterwards,

occlusion advances and the systems fill in and consequently core pressure slowly increases with time. On the other hand, the pressure changes for NEC is typically much smaller (around 10 hPa) and peak intensity is sometimes attained after MDP+12 hours. The increase of core pressure over time after MDP+6h is thus not identifiable for NEC systems.

Regarding the concurrence of these events with ARs, Figure 3 shows the ratios of observed

coincidence between the EC and the presence of an AR within a 1500km radius of the cyclone for the North Atlantic (Fig. 3a) and for North Pacific (Fig. 3b). The first prominent result is high ratios of coincidence between ARs for EC, peaking between 70% and 80% for time lags MDP-6h to MDP+6. When focusing on the North Atlantic region (Fig. 3a), the maximum (~78%) is found for the MDP+6h using the GUAN2016, while using the EIRAS2016 the maximum (~74%) is at the MDP timing. Likewise,

for the Pacific Basin (Fig. 3b) the ratios of coincidence with the ARs reach a maximum of 78% when using the GUAN2016 on the MDP-6, while when using the EIRAS2016 its maximum (~72%) is found at the MDP. The results for the EC are in line with those found by Zhu and Newell (1994) and Ferreira et al. (2016) for a few selected case studies, where ARs were identified near the cyclones during an explosive cyclogenesis. In addition, we show here that the temporal association between the ARs and explosive

deepening of the cyclone takes place primarily between -6h to +6h around the MDP. Our results support the findings by Pirret et al. (2016), as the presence of the ARs will enhance the warm advection ahead of the cyclone core during the development phase.




For NEC, the concurrence of ARs during the development phase is strongly reduced. For the North Atlantic Region (Fig. 3a), the values for NEC range from about 55% with the GUAN2015 and close to 45% with the EIRAS2016 method. For the North Pacific basin (Fig. 3b), results for the NEC are similar (but lower ratios) to those found for the North Atlantic basin ranging from approximately 46% in the

GUAN2015 and nearly 42% when using EIRAS2016.  In addition, there is apparently an increase in the ratio of coincidence between the position of the cyclone and the presence of the ARs from -36h to +36h. This can be associated with the convergence of moisture along the frontal system of the cyclones (Dacre et al., 2015) along with tropical moisture export episodes (Knippertz and Wernli (2010) which can potentiate the formation of an AR in the latter stages of the NEC.

The conclusions attained with both methodologies are very similar. While a clear peak is identified close to the MDP for EC, for NEC a quasi-linear relationship with the ARs is identified in both methodologies however the ratio of coincidence is always higher when using the GUAN2015.

Regarding the spatial distribution of the EC-AR coincidences, no conclusions can be achieved based on our results. Fig. S2 shows the position of the EC during the MDP if the AR coincidence was

detected (red dots) and if it was not (black crosses). At first sight, the coincidences and non-coincidences of explosive cyclonenesis with ARs seem to be roughly equally distributed throughout the Atlantic and Pacific domains. However, on the downstream end of the storm tracks, cases with concurrent AR seem to dominate (e.g. over the British Isles). Still, no general conclusions can be taken on a possible relation on the location of explosive cyclogenesis and concurrence (or non-concurrence) with ARs.

In order to analyse the flux of moisture in near the cyclones within ±36 hours of the MDP, spatial composites of the IVT within radius of 1500km from the cyclone core were computed for each time step, for the EC and for the NEC and for both domains. Fig. 4 shows the composites of the IVT in the surroundings of EC between -36h and +36h from the MDP at 6h time steps for the North Atlantic, while Fig. 5 shows the same fields but for NEC. One key difference between both Figures is noticeable

differences in the IVT fields, which implies the presence of intense IVT values akin to ARs-like structures in explosive cyclogenesis when compared to the NCE events. Since ARs are commonly spatially associated to the warm sector of the cyclone, the evolution of the IVT fields throughout the 36h wind-frame adjacent to the MDP frame depicts the general frontal evolution over the cyclogenesis event. Note



that the AR is already quite prominent at MDP-36h for EC events, and its position slowly rotates around the cyclone core MDP with slightly increasing intensities. Also noteworthy is the fact that after the MDP point the AR not only strongly weakens but its central axis tends to be detached from the cyclone with increasing time, corresponding to the detachment from the warm sector from the cyclone core at later stages of cyclone development (occlusion is initiated). This conclusion is in line with that obtained by Zhu and Newell (1994) for a much smaller number of cases. Figs. S3 and S4 show the composites for EC and NEC events but for the North Pacific domain. No meaningful differences can be observed between the North Atlantic and the North Pacific basins, and thus the conclusions are the same for the North Atlantic. The results suggest the importance of latent heat released when the cyclones encounter the ARs leading to intense condensation process, thus providing an important source of energy when the cyclone is in its deepening phase (e.g. Danard, 1964; Bullock and Jonhson, 1971, Whitaker and Davis, 1993).

The results have revealed two main new insights. First, the highest ratio of the present of the ARs in the vicinity of EC is found within ±6 hours of the MDP. Second, it is apparent for EC events that the AR is located very close to the cyclone centre prior to MDP, while the AR becomes detached from the EC core once the cyclone stops deepening. As a result of this, we confirm the hypothesizes that the presence of an AR raises the odds of an explosive cyclogenesis occurrence and is thus not only a merely a statistical relationship as suggested by Ferreira et al., 2016 for three modelled EC case studies.

## 5 Conclusions

We investigated the importance of ARs in the development of explosive cyclogenesis on both North Atlantic and North Pacific basins using two different algorithms for AR identification. With this aim, the concurrence of the presence of AR in the vicinity of developing cyclones was quantified over different time lags. The main results are summarized in the following:

- ARs are present very frequently within the vicinity of cyclones undergoing EC, reaching a maximum values close to 80% near the MDP (+/- 6 hours) for both domains. The concurrence of ARs with NEC is reduced to 42-46% for NEC.

- While slightly different results are obtained with the two AR methodologies, the results are consistent, both in terms of the general numbers and the time evolution of concurrences





between AR and cyclogenesis over time. While a clear peak is found for EC, a quasi-linear relationship is identified for NEC. The obtained conclusions are thus robust and largely independent of the of the detection ARs algorithm used

- Since ARs are commonly associated to the warm sector of the cyclone, the evolution of the IVT fields throughout the ±36h frame surrounding the MDP point depicts the general frontal evolution over the cyclogenesis event. Prior to the MDP, high values of IVT are already present at -30 hours, with the maximum values of IVT appearing around the MPD (±6h). Afterword's, the IVT values quickly decrease and the central AR axis tends to be detached from the cyclone.

- The analysis of NEC composites reveals much lower values of IVT during development. This is a clear indication of the unusual characteristics of the IVT for EC cases.

The above results strongly indicate that the presence of an AR near the developing cyclone is related with a higher probability of an explosive cyclogenesis occurrence. A detailed analysis of the time evolution of the high values of water vapour flux associated with the AR during the cyclone development phase leads us to hypothesize that this fact is a fingerprint of a physical mechanism that raises the odds of an explosive cyclogenesis occurrence and not merely a statistical relationship. Given the previous work of Zhu and Newell (1994) on selected case studies, our analysis allows for a systemization of results from a climatological perspective. This insight can be potentially helpful to enhance the predictability of high impact weather associated with explosive cyclones and atmospheric rivers.

Regarding future climate projections, Ramos et al. (2016) showed that most models from the CMIP5 project a coherent increase of IVT values over the North Atlantic Basin and an increase in the number of ARs that hit the Western Europe by the end of the XXI century, although more evident with emissions scenario RCP8.5 than with scenario RCP4.5. When such climate change scenarios are considered simultaneously with the new insights of the current paper, this implies that the probability of occurrence of intense extra-tropical explosive cyclones will increase in future decades.



**Author Contributions**

JEB, AMR, JGP developed the concept of the paper and wrote the first manuscript draft. JEB performed the data analysis and prepared the figures. MLR provided the cyclone track data. All authors contributed with ideas, interpretation of the results and manuscript revisions.

**Acknowledgments**

The ECMWF ERA-Interim reanalysis data were obtained from http://www.ecmwf.int/en/research/climate-reanalysis/erainterim. JEB would like to thank Bin Guan for kindly sharing the ARs detection database. JEB was financially supported by the Spanish government (MINECO) and Xunta de Galicia (CGL2013-45932-R, GPC2015/014 - ERDF), and contributions by the

COST action MP1305 and CRETUS Strategic Partnership (AGRUP2015/02). AMR was supported through a postdoctoral grant (SFRH/BPD/84328/2012) from the Portuguese Science Foundation (Fundação para a Ciência e a Tecnologia, FCT). AMR and RMT were supported by the project IMDROFLOOD – Improving Drought and Flood Early Warning, Forecasting and Mitigation using real-time hydroclimatic indicators (WaterJPI/0004/2014) funded by Fundação para a Ciência e a Tecnologia,

Portugal (FCT). JGP thanks AXA Research Fund for support. The authors thank Helen Dacre and Vicente Pérez Muñuzuri for helpful discussions.




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





**Figure Captions**

**Figure 1.** Spatial distribution of the location where explosive cyclogenesis reach their minimum core pressure for: a) Atlantic Ocean and b) Pacific Ocean. Contours correspond to the average number of events per extended winter (ONDJFM) season, detected per 5º x 5º area normalized for 50ºN. The spatial
distribution was smoothed with a 5° averaging radius.

**Figure 2.** Example of a well-defined Atmospheric River associated to an explosive cyclone development land-falling the British Isles on 31 January 1988, 18 UTC. Mean Sea Level Pressure field (hPa) is indicated as black isolines in all panels. (a) total integrated column of water vapor (IWV, colours, kg.m$^{-2}$)
and integrated vapour transport (IVT, arrows, kg.m$^{-1}$.s$^{-1}$). (b) Shape region (red) and central axis of the Atmospheric river (blue) for the GUAN2015 algorithm. (c) as a) but showing only IVW values above 10 kg.m$^{-2}$ and the blue crosses highlights the central axis of the Atmospheric River detected by the EIRAS2016 algorithm. In addition the location of the MDP is highlight with a red dot.

**Figure 3.** (a) Ratio of coincidence between the position of the cyclones for the North Atlantic basin and the presence of an Atmospheric River in a 1500 km radius. The maximum deepening point (MDP) is fixed as time-reference and results are shown for ±36 hours of the MDP. Red lines correspond to the GUAN2015 method and black lines to the EIRAS2016 method. Solid lines refer to explosive cyclogenesis (EC) and dotted lines refer to Non-Explosive events (NEC). (b) as a) but for the North Pacific basin.

**Figure 4.** Composite of the integrated vapour transport (IVT, colours, kg.m$^{-1}$.s$^{-1}$) within a 1500km radius around the cyclone core of an explosive cyclogenesis (EC) cyclone for the North Atlantic basin for the period 1979-2011. The maximum deepening point (MDP) is fixed as time-reference and results are shown for ±36 hours of the MDP.

**Figure 5.** Same as Figure 4, but for Non-Explosive cyclogenesis (NEC).







**Figures**

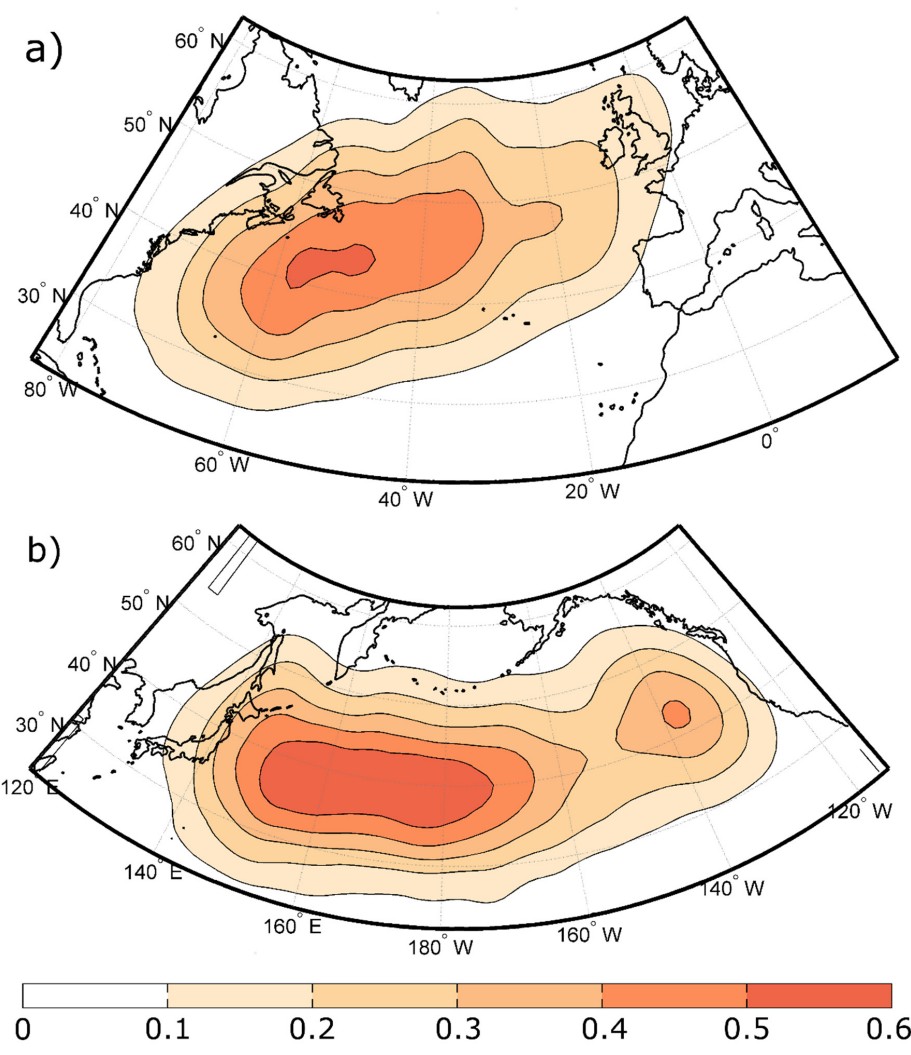

**Figure 1.** Spatial distribution of the location where explosive cyclogenesis reach their minimum core pressure for: a) Atlantic Ocean and b) Pacific Ocean. Contours correspond to the average number of events per extended winter (ONDJFM) season, detected per 5°x5° area normalized for 50ºN. The spatial distribution was smoothed with a 5° averaging radius.





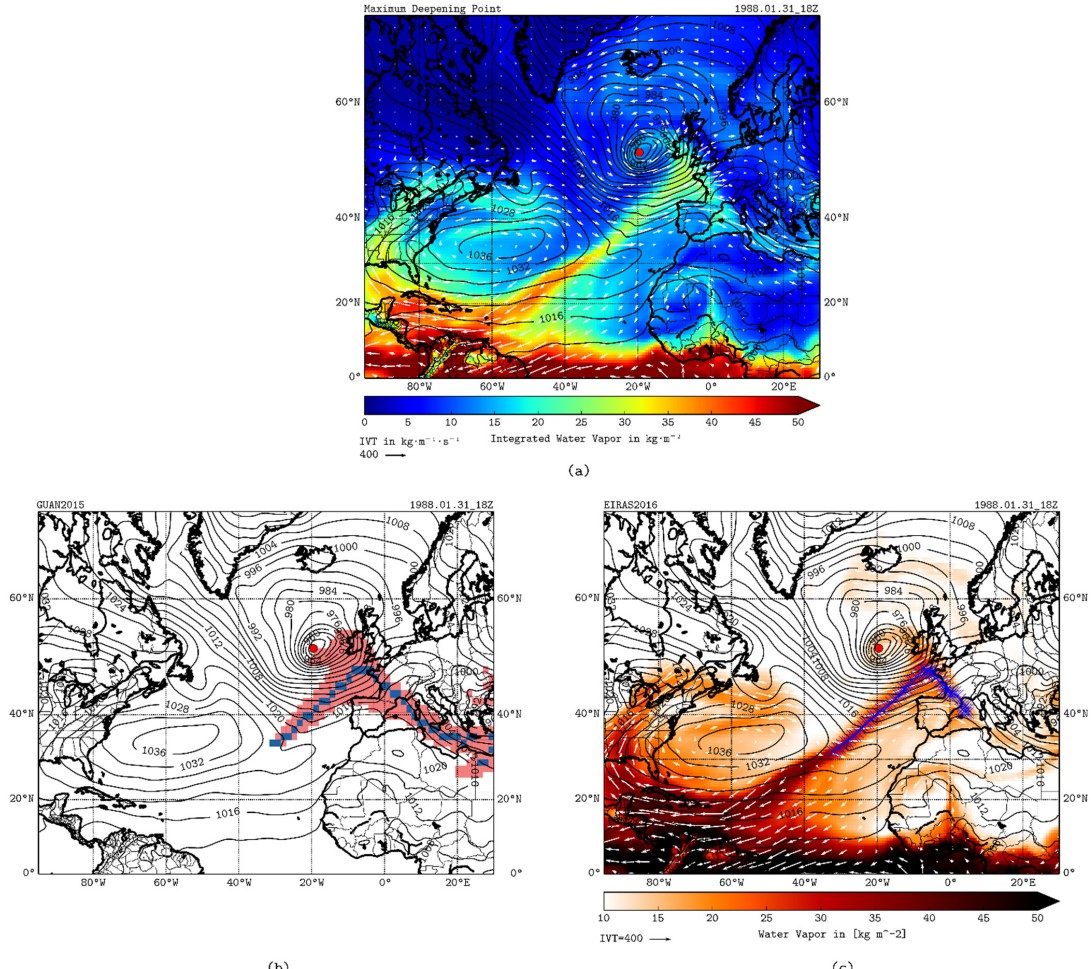

**Figure 2.** Example of a well-defined Atmospheric River associated to an explosive cyclone development land-falling the British Isles on 31 January 1988, 18 UTC. Mean Sea Level Pressure field (hPa) is indicated as black isolines in all panels. (a) total integrated column of water vapor (IWV, colours, kg.m$^{-2}$) and integrated vapour transport (IVT, arrows, kg.m$^{-1}$.s$^{-1}$). (b) Shape region (red) and central axis of the Atmospheric river (blue) for the GUAN2015 algorithm. (c) as a) but showing only IVW values above 10 kg.m$^{-2}$ and the blue crosses highlights the central axis of the Atmospheric River detected by the EIRAS2016 algorithm. In addition the location of the MDP is highlight with a red dot.



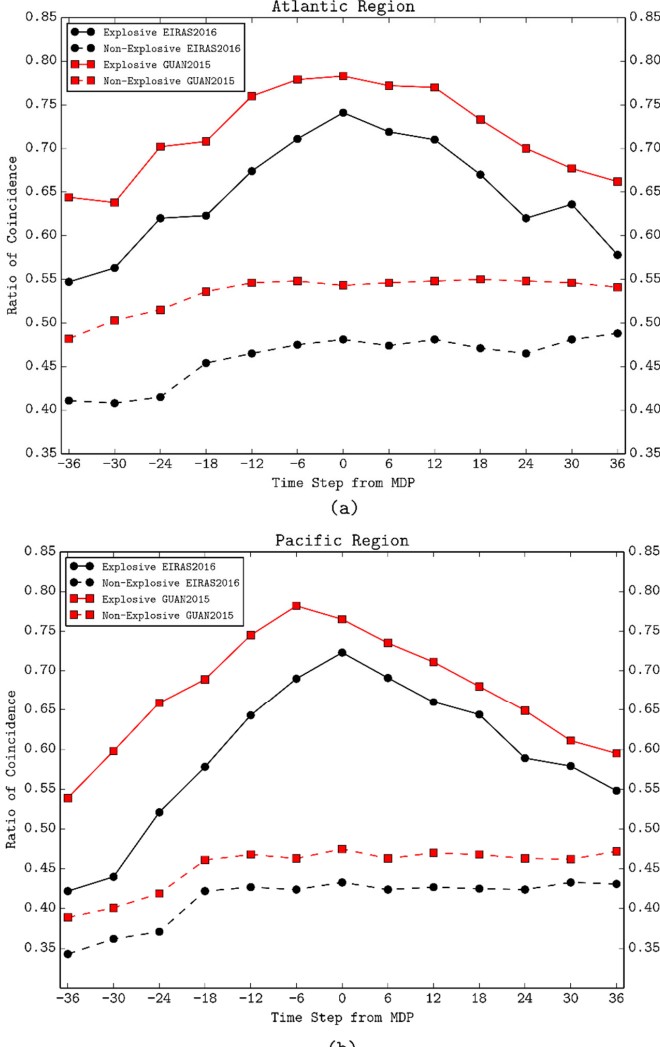

**Figure 3.** (a) Ratio of coincidence between the position of the cyclones for the North Atlantic basin and
the presence of an Atmospheric River in a 1500 km radius. The maximum deepening point (MDP) is
fixed as time-reference and results are shown for ±36 hours of the MDP. Red lines correspond to the
5  GUAN2015 method and black lines to the EIRAS2016 method. Solid lines refer to explosive cyclogenesis
(EC) and dotted lines refer to Non-Explosive events (NEC). (b) as a) but for the North Pacific basin.

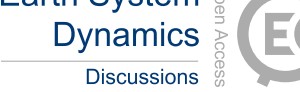



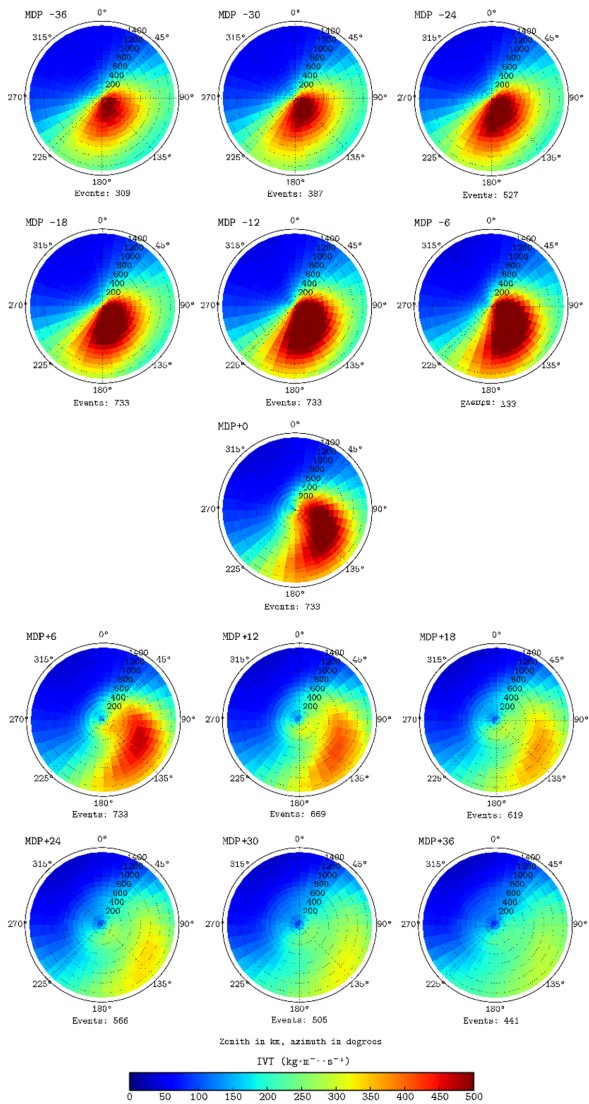

**Figure 4.** Composite of the integrated vapour transport (IVT, colours, kg.m⁻¹.s⁻¹) within a 1500km radius around the cyclone core of an explosive cyclogenesis (EC) cyclone for the North Atlantic basin for the period 1979-2011. The maximum deepening point (MDP) is fixed as time-reference and results are shown for ±36 hours of the MDP.





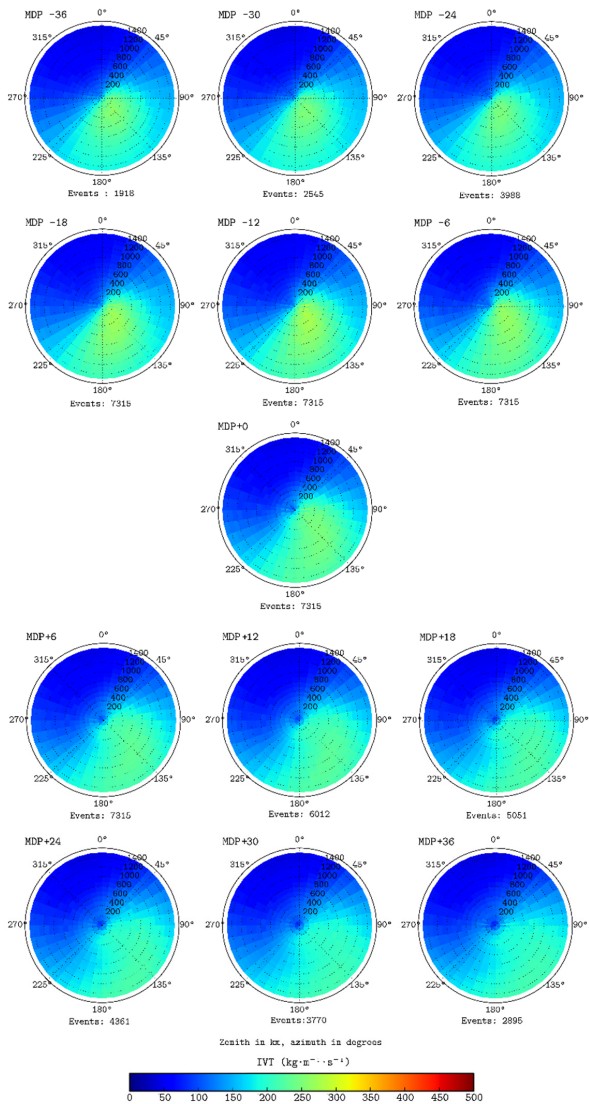

**Figure 5.** Same as Figure 4, but for Non-Explosive cyclogenesis (NEC).