# Peer review of "The concurrence of Atmospheric Rivers and explosive cyclogenesis in the North Atlantic and North Pacific basins"

_Earth System Dynamics, 2017_

## Referee Comment (RC1) · A. Speranza (Referee) · 21 Sep 2017

The concurrence of Atmospheric Rivers and explosive cyclogenesis in the North Atlantic and North Pacific basins Jorge Eiras-Barca, Alexandre M. Ramos, Joaquim G. Pinto, Ricardo M. Trigo, Margarida L.R. Liberato, Gonzalo Miguez-Macho

In this paper, the concurrence of atmospheric rivers and explosive cyclogenesis over the North Atlantic and North Pacific Basins is analysed using ERA-Interim reanalysis data for 1979-2011 (for the extended winter months ONDJFM). Atmospheric rivers are identified in concurrence with almost 80% of explosive deepening cyclones and only in about 40% of the cases of non-explosive cyclones. The Conclusion is offered that "The

above results strongly indicate that the presence of an AR near the developing cyclone is related with a higher probability of an explosive cyclogenesis occurrence. A detailed analysis of the time evolution of the high values of water vapour flux associated with the AR during the cyclone development phase leads us to hypothesize that this fact is a fingerprint of a physical mechanism that raises the odds of an explosive cyclogenesis occurrence and not merely a statistical relationship. ………. This insight can be potentially helpful to enhance the predictability of high impact weather associated with explosive cyclones and atmospheric rivers."

There are some minor errors like, for example: • Pag.6 line 7 "Whereas for the Atlantic storm track has a clear SW-NE orientation is found, reaching values of 0.8 events"; either "has" or "is found" should be omitted; • Pag.4 line 8 "…statistics changes over time (Table S1), as not all systems have the same life time."; "Table" is probably "Figure"; but overall the text is adequately written.

The object of the study is interesting and I believe the analysis can be extended to the role of localized flows of atmospheric water also in other types of adverse weather development. For example, in the analysis of the event which led to the disaster in Giampilieri (October 2009) a concentrated southerly flow of atmospheric water channelled between Sicily and Calabria was the source of intense precipitation which eventually caused the deadly landslide.

However, I have the feeling that the above mentioned conclusions are pushed too far with respect to the real achievements of the analysis reported in the paper: the simultaneous occurrence of different events is in itself no proof of a cause-effect relationship between them and, even less, of a predictive potential. My scepticism is based on personal experience in trying to identify "precursors" of relevant tropospheric developments. Specifically: • in early seventies, following Ed Danielsen (1964,1968,1970), I participated in the search for correlation between tropospheric folding and alpine cyclogenesis(Nanni et al. 1975), but studies on the subject eventually revealed that although stratospheric air enhances signals (due to its very low density) it is too tenuous to exert

any real "forcing" on the troposphere below and, in fact, its analysis does not improve the prediction skill of intense Mediterranean cyclones; • a few years later we went through a similar experience in studies concerning the relationship between stratospheric warming and blocking: sudden stratospheric warming eventually resulted to be the consequence and not the cause of tropospheric blocking.

In conclusion, my feeling is that the conclusive statements of the paper are too generic and I would suggest either to moderate the expectations or be more specific about the physical mechanism alluded to and the associated enhancement in the predictability of high impact weather associated with explosive cyclones.

Danielsen, E. F.: Report on Project Springfield, Headquarters, Defense Atomic Support Agency, Washington D. C. 20301, 15 July 1964.

Danielsen, E. F.: Stratospheric-Tropospheric Exchange Based on Radioactivity, Ozone and Potential Vorticity. J. Atmos. Sci., 25, 502 (1968).

Danielsen, E.F., R. Bleck, J. Shedlovsky, A. Wartburg, P. Haagenson, and W. Pollock: Observed Distribution of Radioactivity, Ozone and Potential Vorticity Associated with Tropopause Folding. J. Geophys. Res., 75, 2353 (1970).

Nanni T., A. Speranza, A. Trevisan and 0. Vittori, 1975: "Precipitation of stratospheric tracers and cyclogenesis in the Western Mediterranean". Arch. Met. Geoph. Biokl., A 24, 321-328.

Antonio Speranza

---

## Referee Comment (RC2) · Anonymous Referee #2 · 21 Sep 2017

The paper is well written and easy to follow, and the subject is relevant. The paper should be published with minor corrections. Minor points follow.

Beginning of page 4: the "historical" flood of Florence in 1966 was also a consequence of an atmospheric river (Malguzzi P., G. Grossi, A. Buzzi, R. Ranzi, R. Buizza: The 1966 "century" flood in Italy: A meteorological and hydrological revisitation. Journal of Geophysical Research: Atmospheres 111, D24, DOI: 10.1029/2006JD007111, 2006 )

Page4 line 20: the formula has a misprint (/ instead of =).

Page 4 line 26: for instead of For.

Page 6 line 7: delete has.

Page 6 line 12: . . .to detect ARs . . ..can broadly be divided into. . .

Page 7 line 7: word should be world.

Page 8 line 12: graphs b) and d) are too flat to identify a minimum.

Page 9 lines 5-8: are you referring to the general trend of NEC cases from -36h to 36h? Please clarify.

Page 9 line 12: . . .methodologies. However,. . ..

Page 9 line 20: . . .moisture flux near the cyclones. . .

Page 10: figures S3 and S4 are really not needed.

Page 10, last paragraph of section 4: hypothesis instead of hypothesized. The evidence that the AR is already present at MPD-36h for EC may be a strong argument to support the conclusion, since it may be considered as a precursor.

Page 11 line 3: of the is repeated.

Page 11, line 18: the sentence "This insight can be potentially helpful to enhance the predictability. . .." is too vague and difficult to be sustained given these results (in my opinion). Please try to be more specific.

---

## Referee Comment (RC3) · Anonymous Referee #3 · 10 Oct 2017

In this article the authors study the relationship between atmospheric rivers and explosive cyclogenesis through the objective identification of both large-scale features. The topic is interesting and within the scope of the journal. However, there are several details in the study and the interpretations of results that require attention before I can fully recommend the article for publication in ESD. These details are listed below in the specific comments followed by a list of purely technical correction.

Specific comments

P1L14: I'm not convinced by the assertion that the occurrence of atmospheric rivers are characteristic features of baroclinic atmospheres. Are there atmospheric rivers on

Mars?

P3L16-24: These lines are taken verbatim from Dettinger et al. (2015). Please rephrase or use quotation marks to indicate that you are using the words already published by another author. (To the editor: I didn't actively look for pieces of text taken without appropriate attribution from other sources.) On the other hand, I don't agree with the clarification on the terms warm conveyor belt (WCB), tropical moisture exports (TME) and ARs made by Dettinger et al. (2015). The main drawback in Dettinger et al.'s clarification is the lack of an explanation as how ARs are formed. In my opinion, they are the footprint of WCBs and possibly other frontal jets, which extract moisture from wetter regions (originally the tropics) to moisten drier regions. From this point of view ARs are a consequence of frontal dynamics. Dettinger et al. state that "[water] vapour is often transported to the WCB by an AR". However, a WCB is an air stream that develops as a consequence of the baroclinic development of a cyclone and frontal structure. Being an air stream, it's the WCB itself the entity that transports the moisture. The moisture would be present or absent depending on whether previous WCBs or other frontal jets transported it. Regarding TMEs, Knippertz and Wernli (2010, doi:10.1175/2009JCLI3333.1) explicitly included what was called AR in the set of TMEs. Therefore, all ARs are TMEs. The authors of the present paper seem to subscribe to this view at times: For instance, in P4L7-10, they seem to use the terms AR and TME as synonyms. All this is not to say that the authors should not be studying ARs. They provide a good definition of ARs (P3L11-13, however see also the comment to P7L20). However, if the authors are willing to enter the debate, this is a good opportunity to provide a better clarification of terms.

P5L16-17: Whether a trigger happens just prior to its effect or long before it is not something that can be guaranteed. Please rephrase.

P7L20: I'm not convinced there is a region with high IVT values extending from the Caribbean to the British Isles. This is precisely where the confusion in the interpretation of ARs arise as it is not IVT, but IWV what extends between these two locations in Fig.

1. Even the two AR-detection methods show that strong IVT is confined in its most southern and western extreme to $30° - 35°$N and about $30°$W, whereas the Caribbean is a long way from this (around $20°$N, $60°$W). Please, rewrite this description.

P9L5-9: Is there really an increase? There is an increase between -36 h and -24 h but after that the lines are essentially flat. The lines in Fig. 3 must include error bars. This might reveal whether the increase is within the error or not. Also, please elaborate on the relationship of this increase and the frontal moisture convergence as it's not clear.

P9L11: What is a quasi-linear relationship? Even if it was a line, I don't see how it helps in the interpretation of results. This term also appears in P11L1.

P9L20-26: This part of the study produced the expected results, which is good, but it can go beyond that. What the composites are showing are the 80

P10L17: I don't see how your description goes beyond a statistical relationship. This is also stated in P11L16. However, to truly remove the statistical character the evolution of whole ARs would need to be studied too so that changes in cyclones can be related to changes in ARs.

Technical comments

P2L18: Delete 'that'.

P2L21: Delete 'the most'. Or how are you measuring the quality of being maritime?

P4L8: Delete 'the' in front of Ferreira et al.

P5L7: Use period instead of colon in $0 : 75° \times 0 : 75°$.

P5L9: It says '...lasting more than 24 hours'. Should it be less, i.e. '... lasting less than. . .'?

P5L13: I don't understand what the authors meant by 'attained'.

P5L14: Use 'rather than' instead of 'over'.

P5L26: 'For' should not start with capital.

P5L27-P6L2: There is no need to give approximate figures. Give the actual percentages.

P6L7: Change 'Whereas' for 'While' and delete 'has'.

P6L12: Delete the second 'be'.

P6L15: Delete 'For' and start the sentence with 'Methods'.

P6L16: Delete ', they'

P6L17: '... combination of IWV and IVT estimated reanalysis datasets' is not clear. Please rewrite.

P6L23: Change 'cf.' to 'see'. Cf. indicates comparison, which is not the case here.

P7L1-2: Plural of radius is radii

P7L4: Delete 'et al.'

P7L4-5: Are Guan and Waliser (2015) also studying ERA-Interim to produce their dataset?

P7L7: Should it say 'world' rather than 'word'?

P8L24: I'm not sure what 'temporal association' means. Change to 'temporal coincidence'.

P9L1: The verb 'reduce' implies that it was once high and now it's low. Perhaps change to 'smaller'.

P9L27-28: What is a 36h wind-frame?

P10L15: Delete 'of this'.

P10L25: Delete 'reduced to'.

P11L7: Should it say '-36 hours' rather than '-30 hours'?

P11L8: Change 'Afterword's' to 'Afterwards'.

―――――――――――――――――――

---

## Author Comment (AC1) · 2 Nov 2017

In this paper, the concurrence of atmospheric rivers and explosive cyclogenesis over the North Atlantic and North Pacific Basins is analysed using ERA-Interim reanalysis data for 1979-2011 (for the extended winter months ONDJFM). Atmospheric rivers are identified in concurrence with almost 80% of explosive deepening cyclones and only in about 40% of the cases of non-explosive cyclones. The Conclusion is offered that "The above results strongly indicate that the presence of an AR near the developing cyclone is related with a higher probability of an explosive cyclogenesis occurrence. A detailed analysis of the time evolution of the high values of water vapour flux associated with the AR during the cyclone development phase leads us to hypothesize that this fact is a fingerprint of a physical mechanism that raises the odds of an explosive cyclogenesis occurrence and not merely a statistical relationship. . . . . . . . . . This insight can be potentially helpful to enhance the predictability of high impact weather associated with explosive cyclones and atmospheric rivers."

Dear Antonio Speranza, thank you for your valued time dedicated to reviewing this paper. We believe that these modifications will improve the manuscript. Here you can find the response to your comments, questions, and suggestions.

There are some minor errors like, for example:
Pag.6 line 7 "Whereas for the Atlantic storm track has a clear SW-NE orientation is found, reaching values of 0.8 events"; either "has" or "is found" should be omitted;
The text was corrected accordingly.

Pag.4 line 8 "…statistics changes over time (Table S1), as not all systems have the same life time."; "Table" is probably "Figure"; but overall the text is adequately written.
In this specific case we are actually refereeing to the Supplementary Table S1. In Table S1, we show the number of explosive cyclones (EC) and non-explosive cyclones (NEC) in each time step used for the computation of Figures 3 to 5 and Supplementary Figures S1, S3 and to S4 for the North Atlantic domain (a) and for the North Pacific domain (b). We agree that the original text was not clear regarding the differences between Supplementary table and figure. Therefore, we have changed the text in the new version of the manuscript in order to highlight when we are referencing a Supplementary Table and when to the Figure.

The object of the study is interesting and I believe the analysis can be extended to the role of localized flows of atmospheric water also in other types of adverse weather development. For example, in the analysis of the event which led to the disaster in Giampilieri (October 2009) a concentrated southerly flow of atmospheric water channeled between Sicily and Calabria was the source of intense precipitation which eventually caused the deadly landslide.
We agree with the reviewer that localized flows of atmospheric water (called Atmospheric Rivers) can lead to huge socio-economic impacts in different regions of the Globe. In fact, the authors contributed to several recent studies analysing extreme precipitation and floods associated with ARs in the Iberian Peninsula

(Ramos et al., 2015; Eiras-Barca et al., 2016, Pereira et al., 2016, Liberato et al. 2013, Trigo et al., 2014). In addition, other studies (as mentioned in the introduction) show the importance of the Atmospheric Rivers in extreme precipitation not only in the west coast of the USA, but also for Europe, including the UK (Lavers et al., 2012), Norway (Sodemann, H. and A. Stohl, 2013) and also another example in Italy as point out by reviewer #2 (Malguzzi et al., 2006). We have included a new reference to support the Italian event Malguzzi et al (2006).

Additional references:
Malguzzi P., G. Grossi, A. Buzzi, R. Ranzi, R. Buizza 2006 The 1966 "century" flood in Italy: A meteorological and hydrological revisitation. Journal of Geophysical Research: Atmospheres 111, D24
Sodemann, H. and A. Stohl, 2013: Moisture Origin and Meridional Transport in Atmospheric Rivers and Their Association with Multiple Cyclones. Mon. Wea. Rev., 141,2850–2868
Pereira, S., Ramos, A. M., Zêzere, J. L., Trigo, R. M., and Vaquero, J. M. (2016) Spatial impact and triggering conditions of the exceptional hydro-geomorphological event of December 1909 in Iberia, Nat. Hazards Earth Syst. Sci., 16, 371-390, https://doi.org/10.5194/nhess-16-371-2016
Trigo, R. M., Varino, F., Ramos, A. M., Valente, M., Zêzere, J. L., Vaquero, J. M., Gouveia, C. M., and Russo, A.: The record precipitation and flood event in Iberia in December 1876: description and synoptic analysis, Front. Earth Sci., 2, 1–15,

However, I have the feeling that the above mentioned conclusions are pushed too far with respect to the real achievements of the analysis reported in the paper: the simultaneous occurrence of different events is in itself no proof of a cause-effect relationship between them and, even less, of a predictive potential.

My scepticism is based on personal experience in trying to identify "precursors" of relevant tropospheric developments. Specifically: in early seventies, following Ed Danielsen (1964,1968,1970), I participated in the search for correlation between tropospheric folding and alpine cyclogenesis (Nanni et al. 1975), but studies on the subject eventually revealed that although stratospheric air enhances signals (due to its very low density) it is too tenuous to exert any real "forcing" on the troposphere below and, in fact, its analysis does not improve the prediction skill of intense Mediterranean cyclones; a few years later we went through a similar experience in studies concerning the relationship between stratospheric warming and blocking: sudden stratospheric warming eventually resulted to be the consequence and not the cause of tropospheric blocking.

In conclusion, my feeling is that the conclusive statements of the paper are too generic and I would suggest either to moderate the expectations or be more specific about the physical mechanism alluded to and the associated enhancement in the predictability of high impact weather associated with explosive cyclones.

This is a fair comment, which we partially agree with. Also following the comments by the other reviewers, we have changed the pertinent text and made an effort to "moderate the expectation" regarding the content of the paper.

Danielsen, E. F.: Report on Project Springfield, Headquarters, Defense Atomic Support

Agency, Washington D. C. 20301, 15 July 1964.

Danielsen, E. F.: Stratospheric-Tropospheric Exchange Based on Radioactivity, Ozone and Potential Vorticity. J. Atmos. Sci., 25, 502 (1968).

Danielsen, E.F., R. Bleck, J. Shedlovsky, A. Wartburg, P. Haagenson, and W. Pollock: Observed Distribution of Radioactivity, Ozone and Potential Vorticity Associated with Tropopause Folding. J. Geophys. Res., 75, 2353 (1970).

Nanni T., A. Speranza, A. Trevisan and 0. Vittori, 1975: "Precipitation of stratospheric tracers and cyclogenesis in the Western Mediterranean". Arch. Met. Geoph. Biokl., A  24, 321-328.

---

## Author Comment (AC2) · 2 Nov 2017

The paper is well written and easy to follow, and the subject is relevant. The paper should be published with minor corrections. Minor points follow.
Thank you for your valued time dedicated to reviewing this paper. We believe that these modifications will improve the manuscript. Here you can find the response to your comments, questions, and suggestions.

Beginning of page 4: the "historical" flood of Florence in 1966 was also a consequence of an atmospheric river (Malguzzi P., G. Grossi, A. Buzzi, R. Ranzi, R. Buizza: The 1966 "century" flood in Italy: A meteorological and hydrological revisitation. Journal of Geophysical Research: Atmospheres 111, D24, DOI: 10.1029/2006JD007111, 2006)
We have included the reference in the text. Thank you very much for point out this very interesting study which we were unaware of.

Page4 line 20: the formula has a misprint (/ instead of =).
We have rephrased the sentence in order to become clear.

Page 4 line 26: for instead of For.
The typo was corrected.

Page 6 line 7: delete has.
The sentence was corrected.

Page 6 line 12: . . .to detect ARs . . .can broadly be divided into. . .
The sentence was corrected.

Page 7 line 7: word should be world.
The typo was corrected.

Page 8 line 12: graphs b) and d) are too flat to identify a minimum.
We agree with the reviewer that it is difficult to identify a minimum pressure after the MDP in Figure S1. We have changed the text accordingly the reviewer's suggestion.

Page 9 lines 5-8: are you referring to the general trend of NEC cases from -36h to 36h? Please clarify.
The sentence was clarified and is in the present form clearly states that the specific lines mentioned by the reviewer are referring to NEC cases.

Page 9 line 12: . . .methodologies. However,. . . .
The sentenced was corrected.

Page 9 line 20: . . .moisture flux near the cyclones. .
The sentence was rephrased.

Page 10: figures S3 and S4 are really not needed.

We understand the reviewer's concern about Supplementary Figures S3 and S4 since the results are similar to the ones found for the Atlantic Basin. However, we believe that its inclusion Supplementary Figures are important for the potential readers outside the North-Atlantic-European domain. For this reason, and since the Supplementary Figures S3 and S4 do not influence the size of the main manuscript we choose to maintain them.

Page 10, last paragraph of section 4: hypothesis instead of hypothesized. The evidence that the AR is already present at MPD-36h for EC may be a strong argument to support the conclusion, since it may be considered as a precursor.

We have replaced hypothesizes by hypothesis.

Page 11 line 3: of the is repeated.

The typo was corrected.

Page 11, line 18: the sentence "This insight can be potentially helpful to enhance the predictability…." is too vague and difficult to be sustained given these results (in my opinion). Please try to be more specific.

This is a fair comment, which we partially agree with. Also following the comments by the other reviewers, we have changed the pertinent text and made an effort to "moderate the expectation" regarding the content of the paper.

---

## Author Comment (AC3) · 3 Nov 2017

In this article the authors study the relationship between atmospheric rivers and explosive cyclogenesis through the objective identification of both large-scale features. The topic is interesting and within the scope of the journal. However, there are several details in the study and the interpretations of results that require attention before I can fully recommend the article for publication in ESD. These details are listed below in the specific comments followed by a list of purely technical correction.

Thank you for your valued time dedicated to reviewing this paper. We believe that these modifications will improve the manuscript. Here you can find the response to your comments, questions, and suggestions.

Specific comments

P1L14: I'm not convinced by the assertion that the occurrence of atmospheric rivers are characteristic features of baroclinic atmospheres. Are there atmospheric rivers on Mars?

We agree with the reviewer that the use of the word "atmospheres" is misleading since we are referring to the Earth atmosphere. Therefore we change the change it to: "The explosive cyclogenesis of extra-tropical cyclones and the occurrence of atmospheric rivers are characteristic features of a baroclinic atmosphere,.........".

P3L16-24: These lines are taken verbatim from Dettinger et al. (2015). Please rephrase or use quotation marks to indicate that you are using the words already published by another author. (To the editor: I didn't actively look for pieces of text taken without appropriate attribution from other sources.)

We are aware that these words were taken verbatim from Dettinger et al. (2015), but we didn't realize that quotation marks were needed, as we provided a clear reference to the author at the beginning of the lines "Recently, some agreement has been achieved (Dettinger et al., 2015) regarding the relationships between ARs, warm conveyor belts (WCBs), and tropical moisture exports (TMEs). The term WCB refers......". And in addition, we repeated the reference to Dettinger et al. (2015) at the end of the paragraph. We will re-phrase the text in the revised manuscript and we apologize for the misunderstanding. There are no other pieces of the text being taken without the appropriate attribution from other sources as proven by the iThenticate.com Similarity Report provided by the Journal.

On the other hand, I don't agree with the clarification on the terms warm conveyor belt (WCB), tropical moisture exports (TME) and ARs made by Dettinger et al. (2015). The main drawback in Dettinger et al.'s clarification is the lack of an explanation as how ARs are formed. In my opinion, they are the footprint of WCBs and possibly other frontal jets, which extract moisture from wetter regions (originally the tropics) to moisten drier regions. From this point of view ARs are a consequence of frontal dynamics. Dettinger et al. state that "[water] vapour is often transported to the WCB by an AR". However, a WCB is an air stream that develops as a consequence of the baroclinic development of a cyclone and frontal structure. Being an air stream, it's the WCB itself the entity that transports the moisture. The moisture would be present or absent depending on whether previous WCBs or other frontal jets transported it.

We fully acknowledge that there is no consensual definition on how atmospheric

rivers relate to WCBs, TMEs, etc, and also on the controversial discussion on how atmospheric rivers are formed. There are some recent works that try to prove the origin of the moisture sources of the ARs by means of Lagrangian analysis:

a) Sodemann and Stohl (2013) showed that in December 2006 several ARs reached from the subtropics to high latitudes, inducing precipitation over western Scandinavia. The sources and transport of water vapour in the North Atlantic storm track during that month were examined, and they reveal that the ARs were composed of a sequence of meridional excursions of water vapour. Different moisture sources were found: (1) in cyclone cores, the rapid turnover of water vapour by evaporation and condensation was identified, leading to a rapid assimilation of water from the underlying ocean surface; (2) in the regions of long-range transport, water vapour tracers from the southern edges of the midlatitudes and subtropics dominated over local contributions.

b) Ramos et al., 2014 showed moisture sources for the major ARs affecting western European coasts between 1979 and 2012 over the winter half-year (October to March). The major climatological areas for the anomalous moisture uptake extend along the subtropical North Atlantic, from the Florida Peninsula (northward of 20º N) to each sink region, with the nearest coast to each sink region always appearing as a local maximum. In addition, during AR events the Atlantic subtropical source is reinforced and displaced, with a slight northward movement of the sources found when the sink region is positioned at higher latitudes. In conclusion, the results confirm not only the anomalous advection of moisture linked to ARs from subtropical ocean areas but also the existence of a tropical source, together with mid latitude anomaly sources (associated with convergence of moisture along the fronts).

c) More recently, Jorge Eiras-Barca et al. 2017, analysed two extreme ARs events by using a 3D Tracer tool coupled to the WRF model. Results show that between 80% and 90% of the moisture advected by the ARs, as well as 70% to 80% of the associated precipitation have a tropical or subtropical origin. Local convergence transport is responsible for the remaining moisture and precipitation.

It may also be useful in this context to emphasise that a new definition of ARs was recently included in the AMS glossary (http://glossary.ametsoc.org/wiki/Atmospheric_river) which states that: ARs are "a long, narrow, and transient corridor of strong horizontal water vapour transport that is typically associated with a low-level jet stream ahead of the cold front of an extratropical cyclone. The water vapour in atmospheric rivers is supplied by tropical and/or extratropical moisture sources. Atmospheric rivers frequently lead to heavy precipitation where they are forced upward—for example, by mountains or by ascent in the warm conveyor belt. Horizontal water vapour transport in the midlatitudes occurs primarily in atmospheric rivers and is focused in the lower troposphere."

Therefore it is appropriate to state that the authors are particularly active (in other works) in trying to understand the origin of the moisture and how it is transported by the ARs to the mid latitudes. Nevertheless we partially agree with the reviewer that the definition provided by Dettinger et al., 2015 can be slightly misleading and further

studies need to be undertaken in order to better understand the connection between the TME and how is advected by the ARs to the mid-latitudes. *However from our point of view this discussion is out of the scope of the present manuscript.*

In the particular paragraph mentioned by the reviewer based on Dettinger et al. (2015) we choose to delete it and rephrase it as follows:
"According to the AMS glossary ARs are: "A long, narrow, and transient corridor of strong horizontal water vapour transport that is typically associated with a low-level jet stream ahead of the cold front of an extratropical cyclone". The definition also affirms that the water vapour in ARs is supplied by sourced of both tropical and/or extratropical origin (e.g. Ramos et al.2016a; Eiras et al., 2017) and that ARs can lead to heavy precipitation whenever these systems are forced upward— either by mountains or by ascent in the warm conveyor belt. Horizontal water vapour transport in the midlatitudes occurs primarily in atmospheric rivers and is focused in the lower troposphere."

Regarding TMEs, Knippertz and Wernli (2010, doi:10.1175/2009JCLI3333.1) explicitly included what was called AR in the set of TMEs. Therefore, all ARs are TMEs. The authors of the present paper seem to subscribe to this view at times: For instance, in P4L7-10, they seem to use the terms AR and TME as synonyms.
The authors do not agree that the term ARs and TME are synonyms. It was our mistake to include TME on that particular sentence therefore we have replaced TME by ARs in this sentence.
From our understanding of the study from Knippertz and Wernli, 2010 you can have TME but not necessarily the formation of ARs.

All this is not to say that the authors should not be studying ARs. They provide a good definition of ARs (P3L11-13, however see also the comment to P7L20). However, if the authors are willing to enter the debate, this is a good opportunity to provide a better clarification of terms.
Please see the previous comments. We agree with reviewer that the exact mechanism leading to how the ARs are formed is still an open topic for some authors however this debate is out of the scope of the present manuscript.

P5L16-17: Whether a trigger happens just prior to its effect or long before it is not something that can be guaranteed. Please rephrase.
We agree with the reviewer that the sentence was misleading, therefore the sentence was re-written.

P7L20: I'm not convinced there is a region with high IVT values extending from the Caribbean to the British Isles. This is precisely where the confusion in the interpretation of ARs arise as it is not IVT, but IWV what extends between these two locations in Fig.1. Even the two AR-detection methods show that strong IVT is confined in its most southern and western extreme to $30° – 35°N$ and about $30°W$, whereas the Caribbean is a long way from this (around $20°N$, $60°W$). Please, rewrite this description.
In this particular example the reviewer is right that the IVT does not extend back till the Caribbean region due to the presence of a high pressure system that was located East of Florida. That high pressure location hampered the supply of water vapour to the ARs

(since the IVT is southwards). Therefore the text was changed accordingly to the following:

"The overall IWV pattern is clearly compatible with the presence of an AR-like structure located in the North Atlantic Ocean, showing an extensive region with high IWV values extending from the Caribbean to the British Isles (Fig. 2a). In this case, the IVT preferred direction along the high IWV region is directed from SW to NW between the sub-tropic and the cyclone center. However it seems that for these particular time steps the supply of water vapor from the tropics is cut off by the presence of high pressure system located East of Florida that make the IVT direction from the sub-tropics to the tropics."

We would like to show an example where we have an ARs clearly visible in the IVT field (values above 400kg/m/s), and thus the water vapour transport from the tropics is documented. This AR struck directly over the Iberian Peninsula for several time steps. Still, ARs do not need to be directly connected from the tropics to the mid-latitudes.

[Figure]

Figure R1. IVT direction (vectors) and intensity (kg/m/s; colour shading) fields at (a) 0000, (b) 0600, (c) 1200, and (d) 1800 UTC 25 December 1995.

P9L5-9: Is there really an increase? There is an increase between -36 h and -24 h but after that the lines are essentially flat. The lines in Fig. 3 must include error bars. This might reveal whether the increase is within the error or not. Also, please elaborate on the relationship of this increase and the frontal moisture convergence as it's not clear.

We agree with the reviewer that Fig. 3 should display also the uncertainty range for each time step. Therefore we will include the *variance* as an error bar for each MDP time step. The new version of the Figure 3 is shown below:

[Figure]

P9L11: What is a quasi-linear relationship? Even if it was a line, I don't see how it helps in the interpretation of results. This term also appears in P11L1.

We agree with the reviewer that the use of the term quasi-linear relationship is not very clear. Therefore, in the new revised section 3 the text was changed to "While a clear peak is identified close to the MDP for EC, for NEC a stable relationship with the ARs is identified in both methodologies with almost no changes of the ratio of coincidence when analysing the different 6h time frames."

In the conclusion section it was changed to "While a clear peak is found for EC, a stable relationship is identified for NEC"

P9L20-26: This part of the study produced the expected results, which is good, but it can go beyond that. What the composites are showing are the 80

We believe that this comment by the reviewer was somehow cut in half. In the pdf of the revision (shown below) the sentence stops abruptly in "….are showing are the 80". Can the reviewer clarify it?

P9L20-26: This part of the study produced the expected results, which is good, but it can go beyond that. What the composites are showing are the 80

P10L17: I don't see how your description goes beyond a statistical relationship. This is also stated in P11L16. However, to truly remove the statistical character the evolution of whole ARs would need to be studied too so that changes in cyclones can be related to changes in ARs.

We agree with the reviewer this would be a very nice idea, but it is in our opinion out of scope for the present study. We aim to do additional analysis along this direction in a follow-up study, namely by using a high resolution RCM to model the effect of introducing (or removing) an AR in the evolution of different EC and NEC events.

Technical comments
P2L18: Delete 'that'.
The word has been deleted.

P2L21: Delete 'the most'. Or how are you measuring the quality of being maritime?
The words have been deleted.

P4L8: Delete 'the' in front of Ferreira et al.
We have deleted it according to the reviewer suggestion.

P5L7: Use period instead of colon in $0:75° \times 0:75°$.
The typo was corrected.

P5L9: It says '...lasting more than 24 hours'. Should it be less, i.e. '... lasting less than. . .'?
The text was corrected according to the reviewer suggestion.

P5L13: I don't understand what the authors meant by 'attained'.
We agree with the reviewer that "attained" was not the best choice of word. We have replace it by "was computed".

P5L14: Use 'rather than' instead of 'over'.
We have changed the text accordingly.

P5L26: 'For' should not start with capital.
The entire sentence was re-written in order to become clear. The new version is as follows: "Two wide domains over both ocean basins have been selected: for the Atlantic domains latitudes between 25ºN and 65ºN and longitudes between 80ºW and 10ºE are considered, while for the Pacific domain considered longitudes span between 120ºE and 105ºW."

P5L27-P6L2: There is no need to give approximate figures. Give the actual percentages.
The text was re-written for clarity.

P6L7: Change 'Whereas' for 'While' and delete 'has'.
The text was changed according to the reviewer's suggestion.

P6L12: Delete the second 'be'.
The "be" was deleted from the text.

P6L15: Delete 'For' and start the sentence with 'Methods'.
We have changed the text accordingly.

P6L16: Delete ', they'
The word was deleted.

P6L17: '… combination of IWV and IVT estimated reanalysis datasets' is not clear. Please rewrite.
The sentence has been re-written for clarity.

P6L23: Change 'cf.' to 'see'. Cf. indicates comparison, which is not the case here.
We have replaced the "cf "to "see".

P7L1-2: Plural of radius is radii
We have changed the text accordingly.

P7L4: Delete 'et al.'
The "et al." was deleted from the text.

P7L4-5: Are Guan and Waliser (2015) also studying ERA-Interim to produce their dataset?
Indeed, Guan and Waliser (2015) also used ERA-Interim in their algorithm. This information has been included in the text.

P7L7: Should it say 'world' rather than 'word'?
The typo was corrected.

P8L24: I'm not sure what 'temporal association' means. Change to 'temporal coincidence'.

We have changed the word according to the reviewer's suggestion.

P9L1: The verb 'reduce' implies that it was once high and now it's low. Perhaps change to 'smaller'.

We have changed the text accordingly.

P9L27-28: What is a 36h wind-frame?

We have replaced "wind-frame" to "time-frame".

P10L15: Delete 'of this'.

We have delete it according to the reviewer's suggestion.

P10L25: Delete 'reduced to'.

We have delete it according to the reviewer's suggestion.

P11L7: Should it say '-36 hours' rather than '-30 hours'?

The typo was corrected.

P11L8: Change 'Afterword's' to 'Afterwards'.

We have changed the text accordingly.

---

## Referee Comment (RC4) · Anonymous Referee #3 · 9 Nov 2017

Looking into the response from the authors to my comments, they've pointed out that one of my comments was incomplete. The complete comment is the following:

P9L20-26: This part of the study produced the expected results, which is good, but it can go beyond that. What the composites are showing are the 80% ECs associated with ARs and the 60% NECs not associated with ARs. What is more interesting is to find out why there are some EC that are not associated with ARs. Why these cyclones still develop explosively? Or why are there NECs that are associated with ARs? Why these cyclone do not develop explosively? These questions could be at least partially addressed by separating the cyclones in four categories, namely EC-AR, EC-nonAR,

[Figure]

NEC-AR, NEC-nonAR, and producing composites for each one of these. Are there noticeable differences between EC-AR and NEC-AR, or between EC-nonAR and NEC-nonAR? Changing this analysis will also require changing the last bullet point in the conclusions (P11L10-11).
* * *

---

## Author Comment (AC4) · 24 Nov 2017

Thank you very much for adding the comment that was incomplete.

Looking into the response from the authors to my comments, they've pointed out that one of my comments was incomplete. The complete comment is the following:
P9L20-26: This part of the study produced the expected results, which is good, but it can go beyond that. What the composites are showing are the 80% ECs associated with ARs and the 60% NECs not associated with ARs.

Figure 4 and Figure 5, are showing the composites for the entire EC and NEC dataset respectively. The reviewer is right to assume that the composites are dominated by the 80% ECs associated with ARs (Figure 4) and the 60% NECs not associated with ARs (Figure 5).

What is more interesting is to find out why there are some EC that are not associated with ARs. Why these cyclones still develop explosively? Or why are there NECs that are associated with ARs? Why these cyclone do not develop explosively? These questions could be at least partially addressed by separating the cyclones in four categories, namely EC-AR, EC-nonAR, NEC-AR, NEC-nonAR, and producing composites for each one of these. Are there noticeable differences between EC-AR and NEC-AR, or between EC-nonAR and NECnonAR? Changing this analysis will also require changing the last bullet point in the conclusions (P11L10-11).

The questions raised by the reviewer are relevant, but difficult to assess in a "climatological framework" such as the one we are developing here. However we agree with the reviewer that such an analysis is a very pertinent suggestion. Therefore, we have decided to try to address partially the problem. With this aim, we have separated the cyclones in four categories, namely EC-AR, EC-nonAR, NEC-AR, NEC-nonAR and produced the composites for each case for both **Atlantic basin** (Figure A.1 and A.2) and **Pacific basin** (not shown, but the results are very similar) using the results obtained with GUAN2015 (results obtained with EIRAS2016 database are again similar). The results presented on Figure A.1 and A.2 are only for the time frames MPD+36, MDP and MDP-36 but they illustrate well the differences between the EC-AR, EC-nonAR (Figura A.1), NEC-AR and NEC-nonAR (Figure A.2).

As could be expected, there is a considerable difference in the composites between EC-AR and EC-nonAR, with high values of IVT being found on the EC-AR and considerable lower values in the case of the EC-nonAR.

[Figure]

**Figure A.1.** Composite of the integrated vapour transport (IVT, colours, kg.m$^{-1}$.s$^{-1}$) within a 1500km radius around the cyclone core for EC-AR (upper panel) and EC-nonAR (lower panel) for the North Atlantic basin for the period 1979-2011 for +36 hours do the MDP, for the MDP and for -36h of the MDP. These results are based on the AR database developed by GUAN2015 (see Figure 3 of our manuscript).

Regarding the NEC-AR, NEC-nonAR (Figure A.2.) the composites reveal that for the NEC-AR the presence of relatively higher IVT when compared with the NEC-nonAR. However when compared with the EC-AR and even with the EC-nonAR (Figure A.1) these values are much lower (EC-AR) or relatively lower (EC-nonAR).

This apparent contradiction when comparing the NEC-AR and the EC-nonAR (since higher values of IVT are found for the EC-nonAR) results in our opinion from two main effects:

1) the detection of the ARs is based on the IVT percentile threshold climatology, with means higher values of IVT are required in the south than in the north (GUAN2015 and EIRAS2016), therefore the position (latitude, longitude) of the bomb will have influence of the value of IVT required in the detection of the ARs.
2) The sample of the cyclones in the NEC-AR and the EC-nonAR is very different, being considerably higher in the NEC-AR than in the EC-nonAR (see Figure 3 and supplementary Table S1), therefore the IVT composite is very smoothed in the NEC-AR.

[Figure]

**Figure A.2.** Composite of the integrated vapour transport (IVT, colours, kg.m$^{-1}$.s$^{-1}$) within a 1500km radius around the cyclone core for NEC-AR (upper panel) and NEC-nonAR (lower panel) for the North Atlantic basin for the period 1979-2011 for +36 hours do the MDP, for the MDP and for -36h of the MDP. These results are based on the AR database developed by GUAN2015 (see Figure 3 of our manuscript).

Taking these into account, when analysing the EC-AR, EC-nonAR, NEC-AR, NEC-nonAR composites, and despite they are some noticeable differences between them, we do not think that they were unexpected. Thus, we do not think that this additional analysis leads to new / different conclusions. Still, we have decided to include a short reference to the differences between EC-AR and EC-nonAR in section 3 and include Figure A1. in the supplementary material (new supplementary Figure S3 . A more detailed analysis is left for future work.

"While the EC samples are dominated by systems associated with an AR (cf. Fig. 3), this is not always the case. In order to evaluate this in more detail, we analysed additional composites by separating the EC cyclones in two categories, namely EC with AR (EC-AR) and without AR (EC-nonAR; Supplementary Fig. S3 for the North Atlantic Ocean). As expected, results show that there is a considerable difference in the composites between EC-AR and EC-nonAR, with high values of IVT identified for EC-AR and considerable lower values for EC-nonAR. The figures for EC-nonAR are more similar to NEC systems (not shown)."